# 5-Hydroxymethylcytosine localizes to enhancer elements and is associated with survival in glioblastoma patients

Kevin C. Johnson[1,2], E. Andres Houseman[3], Jessica E. King[1,2], Katharine M. von Herrmann[1,2], Camilo E. Fadul[4] & Brock C. Christensen[1,2]

Glioblastomas exhibit widespread molecular alterations including a highly distorted epigenome. Here, we resolve genome-wide 5-methylcytosine and 5-hydroxymethylcytosine in glioblastoma through parallel processing of DNA with bisulfite and oxidative bisulfite treatments. We apply a statistical algorithm to estimate 5-methylcytosine, 5-hydroxymethylcytosine and unmethylated proportions from methylation array data. We show that 5-hydroxymethylcytosine is depleted in glioblastoma compared with prefrontal cortex tissue. In addition, the genomic localization of 5-hydroxymethylcytosine in glioblastoma is associated with features of dynamic cell-identity regulation such as tissue-specific transcription and super-enhancers. Annotation of 5-hydroxymethylcytosine genomic distribution reveal significant associations with RNA regulatory processes, immune function, stem cell maintenance and binding sites of transcription factors that drive cellular proliferation. In addition, model-based clustering results indicate that patients with low-5-hydroxymethylcytosine patterns have significantly poorer overall survival. Our results demonstrate that 5-hydroxymethylcytosine patterns are strongly related with transcription, localizes to disease-critical genes and are associated with patient prognosis.

[1] Department of Pharmacology and Toxicology, Geisel School of Medicine at Dartmouth, Lebanon, New Hampshire 03756, USA. [2] Department of Epidemiology, Geisel School of Medicine at Dartmouth, Lebanon, New Hampshire 03756, USA. [3] Department of Biostatistics, College of Public Health and Human Sciences, Oregon State University, Corvallis, Oregon 97331, USA. [4] Department of Neurology, University of Virginia, Charlottesville, Virginia 22908, USA. Correspondence and requests for materials should be addressed to B.C.C. (email: Brock.Christensen@dartmouth.edu).

Glioblastoma is the most common and malignant primary intracranial tumour accounting for ~60% of gliomas in adults[1]. Glioblastomas are high-grade gliomas (World Health Organization grade IV) that invade surrounding brain tissue, quickly develop resistance to therapy and have a median survival of 16–19 months after treatment with surgery, radiation and chemotherapy[2].

Systematic molecular analyses have advanced our understanding of glioblastoma pathobiology through the identification of common mutations, structural-based genetic alterations and dysregulated epigenomes[3–5]. Among these molecular alterations, severe disturbance of the glioblastoma epigenome has been observed, especially perturbations to global DNA 5-methylcytosine (5 mC) patterns[6–9]. Importantly, DNA methylation alterations have been recognized to have a functional impact on glioblastomagenesis, tumour growth, response to therapy and prognosis[3,7,10,11]. Similar to other tumour types, glioblastomas exhibit patterns of genome-wide 5 mC loss and genomic context-specific DNA hypermethylation when compared with healthy nervous tissues[12]. Epigenomic patterns in cancer are often heterogeneous and can be substantially influenced by the presence of genetic alterations. For example, gliomas that carry mutations in the isocitrate dehydrogenase 1 and 2 (IDH1 and IDH2) genes have been shown to harbour a pattern of DNA hypermethylation at certain promoter regions which results in a glioma CpG island methylator phenotype (G-CIMP)[8]. The epigenomic phenotype of G-CIMP tumours has been associated with distinct copy number alterations, molecular subgroups and a survival advantage in patients with glioblastoma[7,8,13].

While the glioblastoma DNA methylome has begun to be described, additional modifications to DNA that modulate normal gene function have been implicated in disease development[14]. Indeed, the ten–eleven translocation (TET) family of proteins have been shown to function as enzymes capable of altering the methylation status of DNA by converting 5 mC to 5-hydroxymethylcytosine (5 hmC)[14]. Emerging evidence has suggested that 5 hmC: localizes to sites of DNA damage, may act as a transient intermediate in the process of 5 mC demethylation, and may serve as a functional epigenetic mark for regulating transcription[15,16]. In cell lines, 5 hmC has been shown to recruit DNA-binding proteins and 5 hmC production has been reported to be essential for glioblastomagenesis[17]. However, the aetiologic, distribution and prognostic significance of 5 hmC levels in glioblastoma remains unclear. Accordingly, a portrait of both 5 mC and 5 hmC patterns at a nucleotide resolution is necessary for a more complete understanding of the cytosine modifications in glioblastoma.

Use of sodium bisulfite (BS) treatment followed by hybridization to the Infinium DNA methylation arrays (that is, 450 K array) is a common method for interrogating DNA methylation at the single base level[5]. However, sample treatment with sodium BS does not allow disambiguation of 5 mC from 5 hmC. Oxidation of 5 hmC to 5-formylcytosine (5 fC) is achievable with potassium perruthenate treatment, and subsequent sodium BS treatment (oxBS) converts 5 fC and cytosine to uracil and ultimately thymine while only 5 mC remains unconverted by the coupled oxBS treatment[18]. As a result, the selective oxidation step enables disambiguation of 5 hmC from 5 mC measurements[19,20]. Recently, it has been demonstrated that use of paired BS and oxBS treatment on the same samples followed by hybridization to the Infinium DNA methylation array reliably permits accurate quantification of 5 hmC and 5 mC[19,20].

Here, we present the characterization of 5 mC and 5 hmC levels from paired BS and oxBS 450 K assays in thirty glioblastomas. To effectively analyze and interpret this data, we develop and apply a novel algorithm, OxyBS[21], and produce the first epigenome-wide characterization of 5 hmC in glioblastoma. We demonstrate that the glioblastoma genome is globally depleted of 5 hmC and its distribution is dependent on genomic context. We also observe an enrichment of 5 hmC at glioblastoma-specific enhancer elements, alternative mRNA splicing events, and localization of 5 hmC to several genes significantly mutated in glioblastoma. In contrast to the more commonly described repressive nature of 5 mC, 5 hmC levels in our cohort are primarily positively associated with gene expression and open chromatin. Finally, we find that pattern-specific loss of 5 hmC is associated with poor clinical outcome in patients with glioblastoma.

## Results

**Estimation of 5 mC and 5 hmC.** We adapted the BS–oxBS technology to DNA from 30 glioblastomas and applied a novel algorithm, OxyBS, to obtain genome-wide 5 hmC and 5 mC estimates[21]. All samples were primary tumours, fresh frozen and IDH1 and IDH2 wild type. Patient demographic, tumour characteristics and survival data are presented in Table 1. Here, we focused on the identification of 5 hmC distribution and abundance to better understand the role of 5 hmC in glioblastoma pathobiology. Our approach to investigate potential functions of 5 hmC included assessment of five aims (Supplementary Fig. 1): (1) to characterize genomic abundance and distribution of 5 hmC and 5 mC, (2) to delineate high 5 hmC CpGs in glioblastoma, (3) to determine functional annotation of regions with high 5 hmC, (4) to assess the impact of 5 hmC on gene expression (5) and to apply a machine learning algorithm to cluster patients by 5 hmC profiles.

**Global content and genomic distribution of 5 hmC and 5 mC.** First, to qualitatively assess the extent of deregulation in the glioblastoma epigenome we compared the total genomic content of 5 hmC among glioblastoma samples with levels of 5 hmC in the prefrontal cortex. Although we were unable to obtain disease-free cortex samples in our own cohort, we accessed publicly available prefrontal cortex data using the same BS–oxBS protocol in an independent population of individuals that presented with no evidence of neurological impairment (GSE74368, n = 5). To avoid limitations due to the presence of multiple cell types (that is, glial and neuronal) in the prefrontal cortex samples we did not compare CpG-specific differences in 5 hmC. However,

**Table 1 | Patient demographics and tumour characteristics.**

| Patient and tumour characteristics | (n = 30) |
|---|---|
| *Age at diagnosis (Years)* | |
| Median | 67 |
| Range | 34–84 |
| *Sex, no (%)* | |
| Male | 18 (60.0%) |
| Female | 12 (40.0%) |
| *Survival (months)* | |
| Median | 9.8 |
| Range | <1-53 |
| *IDH1 mutation status, no (%)* | |
| No | 30 (100%) |
| Yes | 0 (0.0%) |
| *IDH2 mutation status, no (%)* | |
| No | 30 (100%) |
| Yes | 0 (0%) |

to provide a global perspective on 5 hmC differences we examined the overall proportion of 5 hmC in total cytosine content (that is, summed 5 hmC beta-values across all CpGs within each sample divided by total number of CpGs profiled) between the two tissues. We observed an average 3.5 fold reduction ($P = 6.2E-06$, Wilcoxon rank sum test) in the total 5 hmC content in glioblastoma samples when compared with cortex samples (Supplementary Fig. 2A). While all glioblastoma samples exhibited a decreased level of total 5 hmC there was high variability in total 5 hmC loss across tumours. Potential biological explanations that may account for the variability of total 5 hmC include sample differences in cellular proportions and differences in the expression of the TET enzymes. To this end, we estimated putative cellular proportions from DNA methylation data ('Methods' section) using a non-negative matrix factorization approach (RefFreeEWAS) and found that there were stable estimates of tumour purity across samples (Supplementary Fig. 3A). Thus, differences in 5 hmC levels across tumours may be more strongly associated with the molecular alterations across tumours rather than tumour purity (Supplementary Fig. 3B). We next tested whether the overall proportions of 5 hmC in total cytosine content were associated with the expression of TET enzymes and additional epigenetic enzymes (that is, *TET1, TET2, DNMT3A, IDH1, e*tc). Again, we did not observe any significant relationships ($P > 0.05$, Spearman's rho test, Supplementary Table 1) consistent with prior studies[22]. Finally, we note that the levels of 5 hmC were not significantly associated with total 5 mC content (Supplementary Fig. 2B, $P = 0.18$, linear regression). 5 hmC has been implicated in the DNA demethylation pathway and may suggest functionality of 5 hmC outside simply removal of 5 mC.

At the nucleotide level, the dynamic range of 5 hmC detection was between 0.00 and 0.99 on the beta-value scale. Empirical cumulative density plots for both mean 5 hmC and 5 mC across

all subjects revealed near zero 5 hmC levels in a majority of CpGs and revealed the well-established bimodal distribution for 5 mC (Fig. 1a). To assess the site-specific relationship between 5 hmC and 5 mC levels we computed Spearman correlation coefficients for each CpG and observed that moderate negative correlations exist for a large majority of CpGs, and that <20% of CpGs demonstrated weak positive correlations between 5 hmC and 5 mC (Fig. 1b). There are known differences in the distribution of DNA methylation based upon by genomic context[23]. The dependency of 5 mC levels upon CpG density was observed in glioblastoma and the CpG islands and CpG island shores exhibited demonstrably lower levels of 5 mC (Fig. 1c). To determine whether potential CpG density specific patterns also exist for 5 hmC we next computed the proportions of CpG-specific mean 5 hmC across the four strata of CpG Island, Shores, Shelves and Ocean. Figure 1d displays the distributions of mean 5 mC and 5 hmC as well as statistical assessment over each of the CpG strata. Generally, the patterns of 5 hmC levels were similar to 5 mC and the lowest levels of 5 hmC were found across CpG islands and CpG Island shores. Similar to observations in 5 mC, CpG Island shelves and Ocean regions also harboured the highest levels of 5 hmC. The role of 5 mC in regulating gene expression has best been described for promoter regions and it is known that TET1, which catalyzes the oxidation of 5 mC to 5 hmC, is enriched for binding to DNA at CpG island promoters[24]. To examine the overall distribution of mean 5 mC and 5 hmC across the promoter regions for both cortex tissue and glioblastomas we averaged the cytosine modifications over a four kilobase pair window near transcriptional start sites[23]. We noted the occurrence of hypermethylation (5 mC) across glioblastoma promoter regions and a consistently lower level of glioblastoma 5 hmC proximal to promoter regions when compared with healthy tissue (Fig. 2a,b).

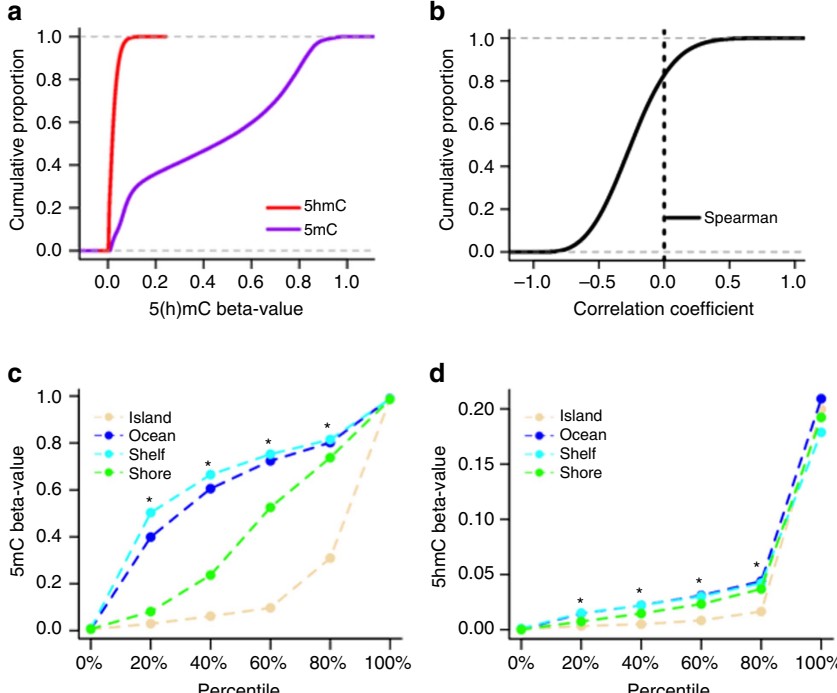

**Figure 1 | 5hmC is depleted and uniquely distributed in glioblastoma. (a)** Empirical cumulative distribution of mean 5-hydroxymethylcytosine (5hmC) and 5-methylcytosine (5mC) across thirty glioblastomas. **(b)** Cumulative proportions of Spearman correlation coefficients calculated for CpG-specific across thirty glioblastomas. **(c)** Percentiles of mean 5mC beta-values for glioblastomas ($n = 30$) across CpG Island strata. Percentiles (that is, quintiles) shown were selected arbitrarily to highlight a large range of values and significant differences between beta-values across the genomic regions at these percentiles were evaluated by Kruskal-Wallis hypothesis tests. Statistical significance ($P < 8.3E-03$, Bonferroni adjusted alpha) is indicated with a *. **(d)** Percentiles of mean 5hmC in glioblastomas ($n = 30$) across CpG Island strata with statistical assessment via Kruskal-Wallis tests.

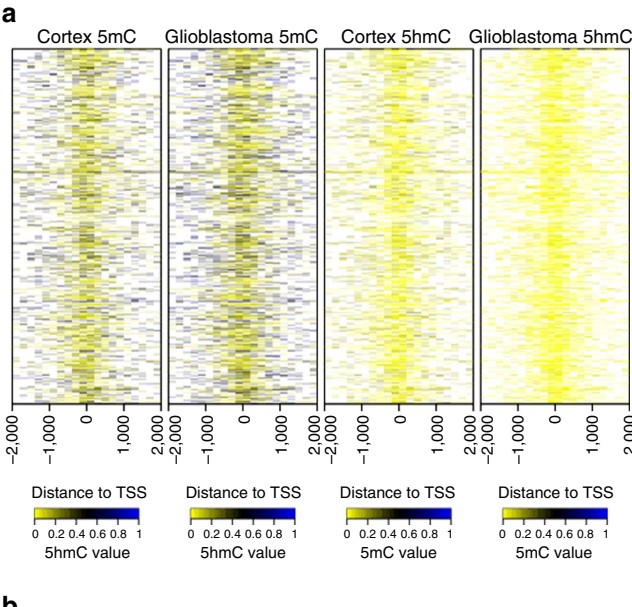

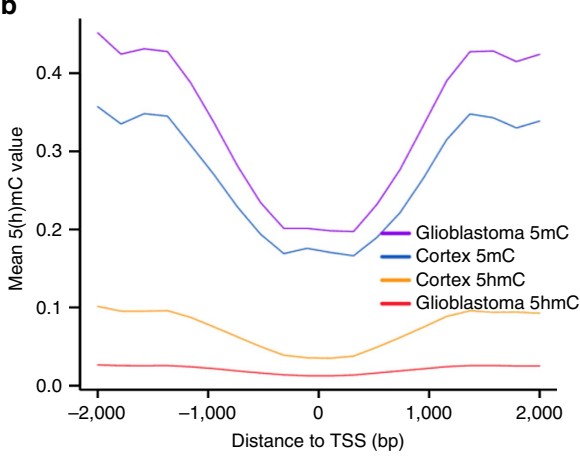

**Figure 2 | Promoter region 5hmC and 5mC in cortex and glioblastoma.**
(**a**) Heat maps represent 5hmC and 5mC levels for 450K array CpGs across 4,000 base pair window in glioblastoma ($n = 30$) and prefrontal cortex ($n = 5$) promoter regions. (**b**) Mean 5hmC and 5mC averages with $+/- 2$ kilo base pairs around canonical transcriptional start sites for glioblastoma ($n = 30$) and prefrontal cortex ($n = 5$).

**5hmC is uniquely distributed in the glioblastoma genome.** To determine which 5hmC sites had the highest potential functional relevance we calculated the mean 5hmC for each CpG across all thirty tumours to identify CpG sites with the highest proportion of 5hmC alleles (Supplementary Fig. 4). In glioblastoma, a large number of CpGs demonstrated mean 5hmC values near zero while signals $>5\%$ 5hmC were found in $\sim 35,000$ CpGs (Supplementary Fig. 4). However, to be confident in distinguishing signal from background we then defined CpGs as high 5hmC CpGs if they were in the highest 1% of mean 5hmC level; resulting in 3,876 CpGs with a mean value of at least $\sim 9.0\%$ 5hmC that we described here as 'high 5hmC CpGs' and are used in subsequent analyses. Among high 5hmC CpGs, 60% were recurrent across a majority of the thirty glioblastomas (2,347/3,876 CpGs with at least 15 tumours displaying $\sim 9\%$ 5hmC at a given CpG) while 30% (1,162/3,876 CpGs) were also among the 1% most variable 5hmC probes suggesting greater inter-individual variability at these locations. The complete list of the high 5hmC CpGs with genomic location, CpG-specific mean

and s.d. is provided as a resource in Supplementary Data 1. The dynamic regulation of DNA methylation for cell-identity is a balance between methylation and demethylation[25]. Here, we visualized the CpG-specific relationships between high 5hmC and 5mC at these high 5hmC CpGs (Fig. 3a) and observed that the highest levels of 5hmC track to sites with intermediate levels of 5mC. These results are consistent with the notion that 5hmC may function as an intermediate in the DNA demethylation process at dynamically regulated regions. Together, the observed loss of 5hmC that occurs in glioblastoma may also reflect disruption of regulated genomic DNA methylation patterns.

Previous sequencing studies in non-diseased tissue have shown that 5hmC marks tend to reside in intronic and gene regulatory regions[26,27]. However, the localization of 5hmC in the glioblastoma genome remains obscure and may influence disease-critical gene expression programs. The genomic distribution of high 5hmC CpGs relative to canonical transcriptional start sites (TSS) is presented in Fig. 3b and reveals that $\sim 75\%$ of high 5hmC were found beyond the 5 kb region upstream of the TSS. To determine whether elevated levels of 5hmC were associated with genomic features, we used a Cochran–Mantel–Haenszel test with binary outcomes (for example, genomic region of interest versus other) to provide a measure of enrichment that permitted stratification by CpG density (that is, CpG islands, shores, shelves and ocean). We observed that high 5hmC CpGs were more likely to be found in intronic regions (2.12 odds ratio (OR), $P = 1.7\text{E-}112$, Cochran–Mantel–Haenszel test, Fig. 3c) as well as depleted in promoter regions (OR $= 0.69$, $P = 4.8\text{E-}21$, Cochran–Mantel–Haenszel test, Fig. 3c) when the genomic regions of the 450 K array were used as a background. These findings concur with previous observations in non-diseased tissue[28,29]. Leveraging publicly available histone H3 lysine 27 acetylation (H3K27ac) genome-wide maps, which marks active enhancers, from three primary glioblastomas we observed substantial enrichment for enhancer regions among the high 5hmC CpGs for all three tumours (median OR $= 3.1$, $P = 1.9\text{E-}211$, Cochran–Mantel–Haenszel test, Fig. 3c). Interestingly, we also found a significant enrichment of high 5hmC CpGs at glioblastoma cell line (U87) defined enhancers (OR $= 2.2$, $P = 1.7\text{E-}46$, Cochran–Mantel–Haenszel test, Fig. 3c) and super-enhancers (OR $= 3.00$, $P = 9.7\text{E-}63$, Cochran–Mantel–Haenszel test, Fig. 3c). Super-enhancers are a subset of enhancers that have critical functions in defining cell-identity and are frequently found at key oncogenic drivers providing support that 5hmC may play key roles in disease[30]. We also noted that there was a modest enrichment among high 5hmC CpGs for glioblastoma DNase hypersensitivity sites, a marker of open chromatin (OR $= 1.32$, $P = 5.34\text{E-}7$, Cochran–Mantel–Haenszel test, Fig. 3c). Furthermore, co-localization of 5hmC with Polycomb repressive complex 2 has been established in embryonic stem cells, but has not been observed in differentiated cells[31]. Similarly, we found no association between 5hmC sites and enrichment for Polycomb group protein target genes in glioblastomas (OR $= 0.96$, $P = 0.48$, Cochran–Mantel–Haenszel test, Fig. 3c)[31].

**Enrichment of genomic regions among critical gene sets.** To provide a broader interpretation of 5hmC function we next sought to identify enrichment of high 5hmC within specific gene sets. To this end, we used the genomic coordinates of high 5hmC CpGs as a query set of regions and tested for enrichment against the background of all CpGs present on the 450 K array in a Genomic Regions Enrichment of Annotations Tool (GREAT) analysis. The top twenty most highly significant gene sets ranked by fold enrichment are presented in Table 2. The high 5hmC

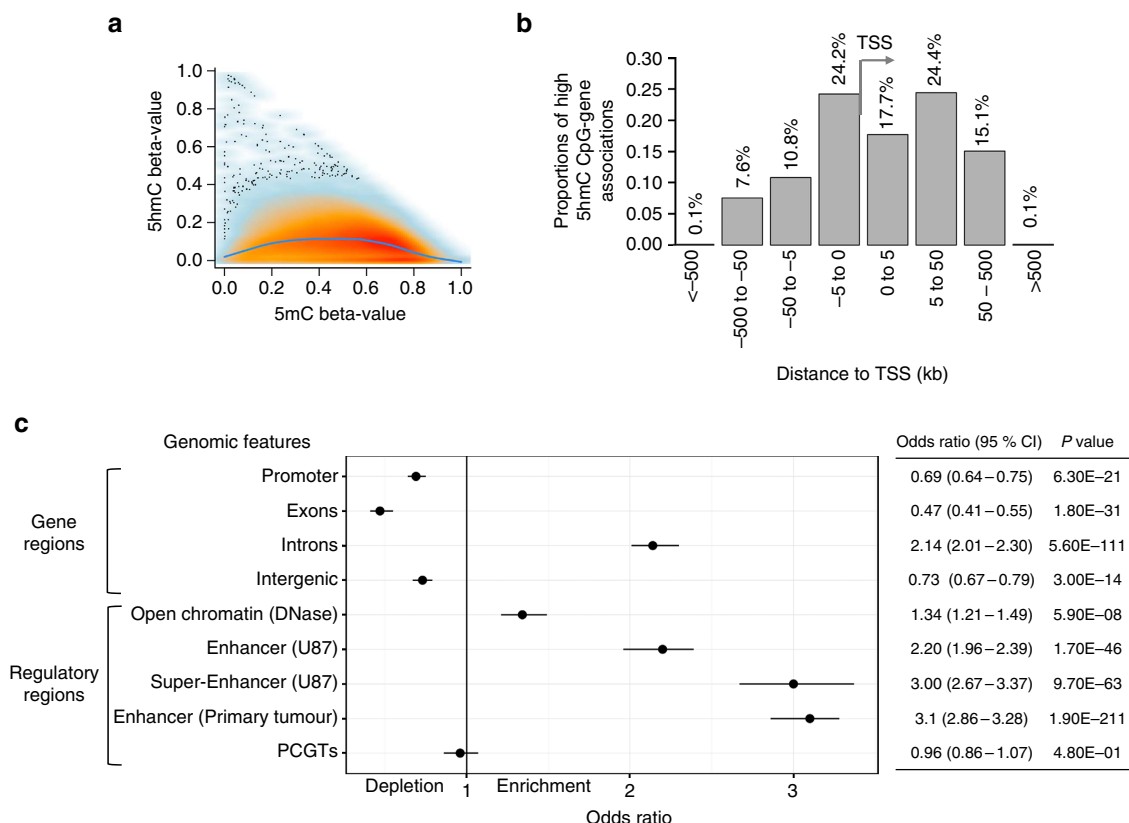

**Figure 3 | 5hmC in glioblastoma is enriched at gene regulatory regions.** (**a**) 5hmC-5mC scatterplot for the highest 5hmC CpGs ($n=3,876$) with smooth curve fitted by loess across thirty glioblastomas. Each point represents a single CpG per tumour with the highest intensity values represented in red and lower intensity (that is, fewer signals) represented in blue. (**b**) The distribution of high 5hmC CpGs relative to the nearest canonical transcriptional start site (TSS) in kilo base pairs with category bins for genomic distance shown for both upstream and downstream of the TSS. Percentage of high 5hmC CpGs ($n=3,876$) are shown above each proportion. (**c**) Forest plot of odds ratios and 95% confidence intervals (Cochran-Mantel-Haenszel or Fisher's exact test) for enrichment of high 5hmC genomic regions against 450K background set. Numerical representation of the odds ratio and associated *P*-value for each genomic feature are also presented.

**Table 2 | GREAT functional enrichment analysis of genomic regions for high 5 hmC CpGs ($n = 3,876$).**

| GO: biological process | Hyper fold enrichment | Hyper FDR Q value |
|---|---|---|
| Negative regulation of receptor catabolic process | 12.37 | 1.81E − 08 |
| Regulation of cellular ketone metabolic process by regulation of transcription from RNA polymerase II promoter | 4.94 | 1.66E − 10 |
| Positive regulation of cardiac muscle hypertrophy | 4.73 | 1.84E − 08 |
| Cytokine production involved in immune response | 4.72 | 9.42E − 09 |
| Regulation of nuclear-transcribed mRNA catabolic process, deadenylation-dependent decay | 4.05 | 3.25E − 10 |
| Positive regulation of nuclear-transcribed mRNA catabolic process, deadenylation-dependent decay | 3.93 | 4.65E − 09 |
| Viral genome replication | 3.82 | 2.85E − 08 |
| Positive regulation of mRNA catabolic process | 3.72 | 3.21E − 09 |
| Regulation of histone deacetylation | 3.45 | 3.11E − 10 |
| Regulation of protein deacetylation | 3.17 | 6.43E − 10 |
| Production of molecular mediator of immune response | 3.14 | 6.15E − 12 |
| Regulation of mRNA stability | 3.13 | 2.51E − 11 |
| Regulation of RNA stability | 3.1 | 1.65E − 12 |
| Peptidyl-threonine modification | 2.92 | 1.71E − 08 |
| Somatic stem cell maintenance | 2.25 | 4.14E − 09 |
| Regulation of fat cell differentiation | 2.24 | 7.68E − 11 |
| Protein import into nucleus | 2.11 | 2.68E − 09 |
| Notch signalling pathway | 2.09 | 9.38E − 13 |
| Nuclear import | 2.08 | 4.12E − 09 |
| Positive regulation of response to external stimulus | 2.05 | 1.72E − 11 |

FDR, False discovery rate; GO, Gene ontology.

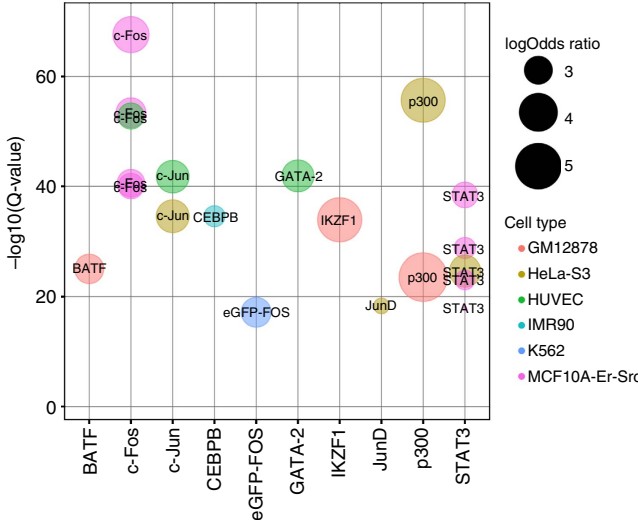

**Figure 4 | Biological interpretation of genomic regions with high 5hmC.** Significance of overlap between high 5hmC regions in glioblastoma with binding sites of transcription factors profiled by ENCODE. The top 10 enriched transcription factors (TFs) as obtained by LOLA analysis are shown. TF are plotted on the x-axis sorted by TF Q-values (Fisher's exact test, corrected for multiple hypotheses testing of 689 TFs) on the y-axis. The log odds ratio for each TF is represented by bubble size and the cell line in which the ChIP-seq experiment was conducted is indicated by bubble colour.

CpGs overlapped significantly with diverse biological pathways including regulation of RNA catabolism and stability, immune response and somatic stem cell maintenance. Enrichment of RNA stability gene sets may reflect shared biological processes of 5hmC between healthy cells and cancer cells as a prior report on adult liver 5hmC made similar observations[26]. In contrast, it is possible that gene sets involved in the immune response and stem cell maintenance are driven by the presence of tumour infiltrating lymphocytes. To validate our pathway enrichment findings we applied an agnostic consensus clustering approach to biological pathways defined in the Kyoto Encyclopedia of Genes and Genomes (KEGG), which considered all 5hmC CpGs on the array. We observed that gene sets involved with metabolism, immunity and RNA processing functions displayed the highest levels of 5hmC (Supplementary Fig. 5A–C).

It has previously been observed that 5hmC is located near transcription factor binding sites (TFBSs) in the mammalian genome[32]. To identify whether particular transcription factors are associated with patterns of high 5hmC we tested for enrichment in binding sites from 689 transcription factor experiments (ENCODE) using the Locus Overlap Analysis (LOLA) software[33]. Relative to genomic regions of the 450 K array, high 5hmC sites were significantly associated with sixty-nine TFBSs ($Q$-value < 0.05, Fisher's exact test, Supplementary Data 2). The top ranked TFBSs enrichments are ranked and the original cell line in which they were profiled is presented in Fig. 4 and the complete rankings can be found in Supplementary Data 2. Several of these transcription factors have established roles in oncogenesis including c-Fos and c-Jun as regulators of cell proliferation and survival, whereas the co-activator p300 is typically found at enhancer regions[34,35]

**5hmC localizes to genes actively transcribed in glioblastoma.** The robust enrichment for 5hmC sites in super-enhancers and

binding sites of proliferation-associated TFs suggests that 5hmC is strongly associated with regulation of disease-specific gene expression programs. To confirm that 5hmC marks are generally associated with active expression in glioblastoma we interrogated the entire glioblastoma transcriptome using RNA sequencing expression levels from The Cancer Genome Atlas (TCGA) ($n = 172$)[13]. Genes were then segregated into the three groups of lowly, moderately or highly expressed genes based on expression tertiles and we tested enrichment for 5hmC sites against the background of the 450 K array. A significant enrichment was found for genes that are actively transcribed in glioblastomas ($P = 5.2E-139$, $\chi^2$ test, Supplementary Table 2). Prior evidence has suggested a role for 5hmC in the regulation of alternative transcript splicing in normal tissues[28]. To examine whether a similar process occurs in cancer, we leveraged the TCGASpliceSeq database for glioblastoma-specific alterations in mRNA splicing patterns of RNAseq data[36]. We found that 5hmC is significantly enriched for exon skip ($OR = 2.03$, $P = 2.23E-48$, Fisher's exact test, Supplementary Table 3) and alternate promoter events ($OR = 2.06$, $P = 9.75E-35$, Fisher's exact test, Supplementary Table 3), but observed neither an enrichment nor depletion in retained introns ($OR = 0.95$, $P = 6.3E-01$, Fisher's exact test, Supplementary Table 3) or alternate donor sites ($OR = 1.12$, Fisher's exact test, Supplementary Table 3).

We next investigated whether gene expression of candidate genes, within our own cohort, were associated with high 5hmC ('Methods' section) using the NanoString nCounter technology. For our candidate gene approach we selected 14 genes with a high proportion of high 5hmC sites in a given gene region (Supplementary Data 3) and a gene with an established relationship to glioblastoma survival, *MGMT* (Supplementary Data 4). In general, 5hmC sites exhibited positive associations with gene expression in our data set (Fig. 5a). In contrast, 5mC at these locations demonstrated primarily negative associations (Fig. 5b). *MGMT* methylation has been used as a biomarker of response to alkylating agents in glioblastoma patients[11]. We found that 5hmC was not associated with *MGMT* expression, but confirmed the strength of association between 5mC and *MGMT* expression (Supplementary Fig. 6). Overall, the strongest association between 5hmC and gene expression among the candidate genes was found in the TSS1500 region of the long non-coding RNA *SOX2-OT* (Spearman correlation coefficient = 0.62, Fig. 5c,d, Supplementary Data 5). Notably, there was no significant association between 5mC and gene expression at this genomic location (Spearman correlation coefficient = −0.18, Fig. 5e). Finally, given our observation that 5hmC was associated with splicing patterns from the TCGA we next assessed whether CpG-specific 5hmC was associated with the differential expression of gene transcript variants from seven genes with high 5hmC in gene regulatory regions (Supplementary Data 4). In our candidate list, CpGs did not exhibit significant associations between differential expression of transcript variants and 5hmC levels (Supplementary Data 6). Taken together, we conclude that 5hmC is often positively correlated with expression and may, in specific gene contexts, be associated with alternative mRNA transcript splicing.

**5hmC profiles are associated with patient survival.** Prior publications have identified an association between decreased total 5hmC (determined by different methods) and poorer survival in glioblastoma patients[37]. Although the tumours we profiled were *IDH1* and *IDH2* wild type, we confirmed that one sample was G-CIMP+ as defined in Noushmehr *et al.*[8] (Supplementary Fig. 7). Patients with G-CIMP+ tumours have a younger age at diagnosis and significantly improved survival (independent of age)[10]. As a result, this sample was excluded from our survival analyses. To investigate the relation of 5hmC

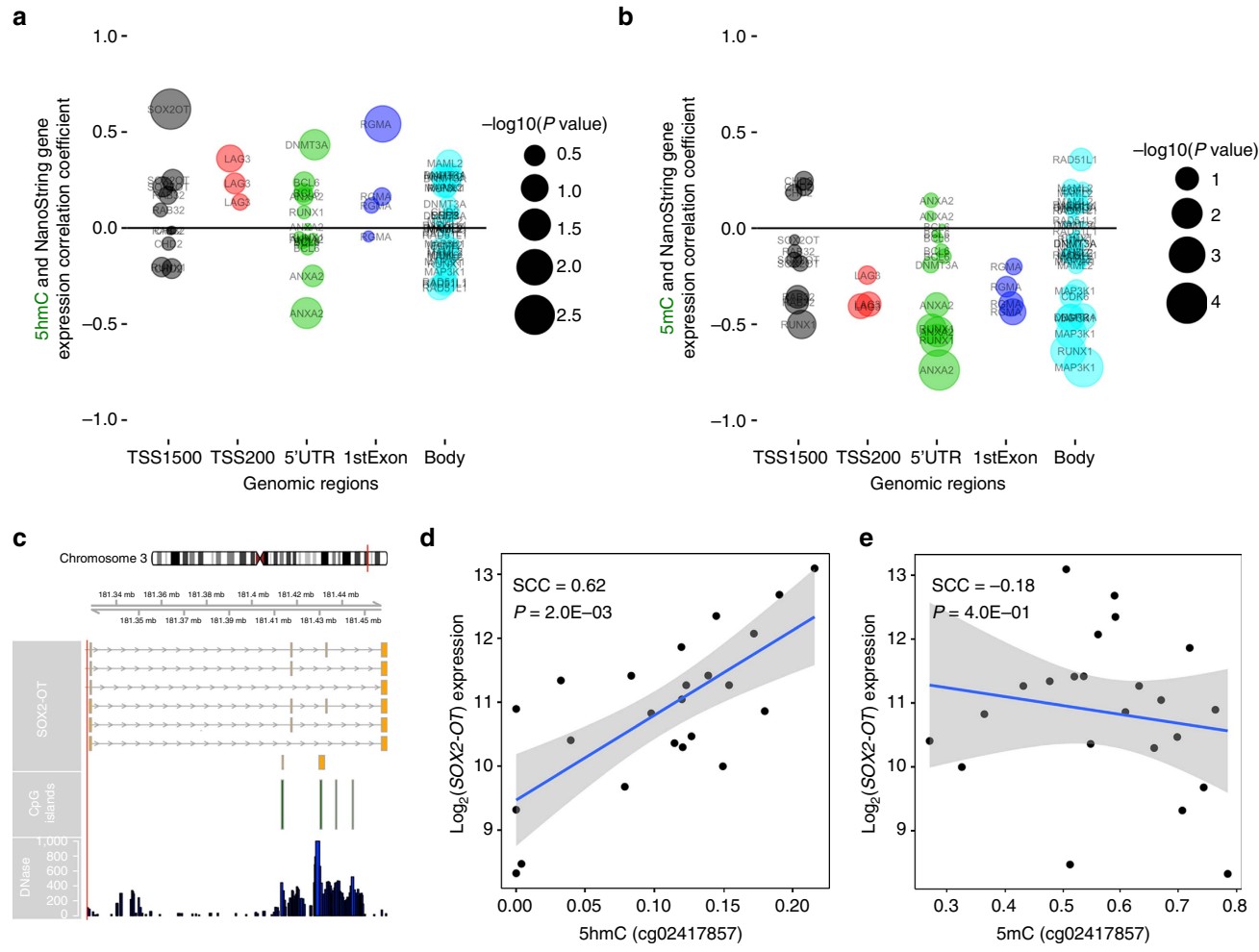

**Figure 5 | 5hmC is positively associated with gene expression.** (**a**) 5hmC sites (that is, CpG position) relative to gene feature are plotted against correlation coefficient derived from Spearman correlations of CpG-specific 5hmC and gene expression. The size of each bubble point represents the nominal statistical significance of the given Spearman correlation with an increasing bubble size corresponding to a smaller *P*-value. The colour of each bubble refers to a distinct gene region. All numerical 5hmC-gene expression correlation coefficients and *P*-values are presented in Supplementary Data 5. (**b**) For each 5hmC site, the correlation between 5mC and candidate gene expression was plotted. Bubble point size also reflects the increasing strength of association based on Spearman correlation. All numerical 5mC-gene expression correlation coefficients and *P*-values are also presented in Supplementary Data 5. (**c**) The genomic location and context of the CpG (cg02417857, 5hmC value) most significantly associated with gene expression mapped to within 1,500 bp of *SOX2-OT* transcription start site. Alternative transcripts, CpG island locations, and DNase hypersensitivity sites from the UCSC genome browser are presented. (**d**) 5hmC levels at cg02417857 were significantly positively correlated with *SOX2-OT* expression (*n* = 23, *P* = 0.002). (**e**) 5mC levels at cg02417857 were not significantly associated with *SOX2-OT* expression (*n* = 23, *P* = 0.40).

patterns (that is, not total 5 hmC) with survival we used a model-based clustering method, Recursively Partitioned Mixture Model (RPMM). RPMM has been extensively used for clustering DNA methylation data to identify classes of tumours based upon 5 mC values, including TCGA[38–40]. Here we applied RPMM to the 3,876 high 5 hmC CpGs and the resulting clustering solution contained two distinct clusters, defining a low and a high 5 hmC cluster (Fig. 6a). Separate clustering analyses were performed for the 1,000, 2,000, 3,000 and 4,000 highest 5 hmC CpGs to examine classification sensitivity and we observed complete stability of cluster membership (that is, samples remained in either low or high 5 hmC clusters regardless of CpG number selected). Cluster membership was associated with total 5 hmC amount (*P* = 2.0E-04, Kruskal–Wallis rank sum test), and patient age (*P* = 0.03, Kruskal–Wallis rank sum test), but not copy number alterations (*P* > 0.05, Fisher's exact test, Supplementary Fig. 8). The cluster of patients with low-5 hmC tumours had an older age at diagnosis (median age = 74.3 years) compared with the high 5 hmC cluster (median age = 65.2 years) and had a shorter median survival (median overall survival = 2.2 months) when compared with the high 5 hmC cluster (median overall survival = 14.5 months). A multivariable Cox proportional hazards model adjusting for age at diagnosis and patient sex resulted in a significantly increased hazard of death related with membership in the low-5 hmC cluster independent of age (Hazard Ratio = 3.3, CI 95% 1.3–8.2, *P* = 0.03, Cox proportional hazards regression, Fig. 6b,c). To compare with prior work that has measured total 5 hmC, we also constructed an index of total 5 hmC by taking the mean across all CpGs considered in the present analysis (*n* = 387,617) and segregated subjects into low and high-total 5 hmC groups based on whether total 5 hmC levels were above or below the median. In a multivariable Cox proportional hazards model adjusted for age at diagnosis and sex, the low total 5 hmC group was not significantly associated with survival (HR = 0.94 CI 95% (0.43–2.03), *P* = 0.88, Cox proportional hazards regression) suggesting that the genomic location of 5 hmC is an important consideration in associations with disease progression.

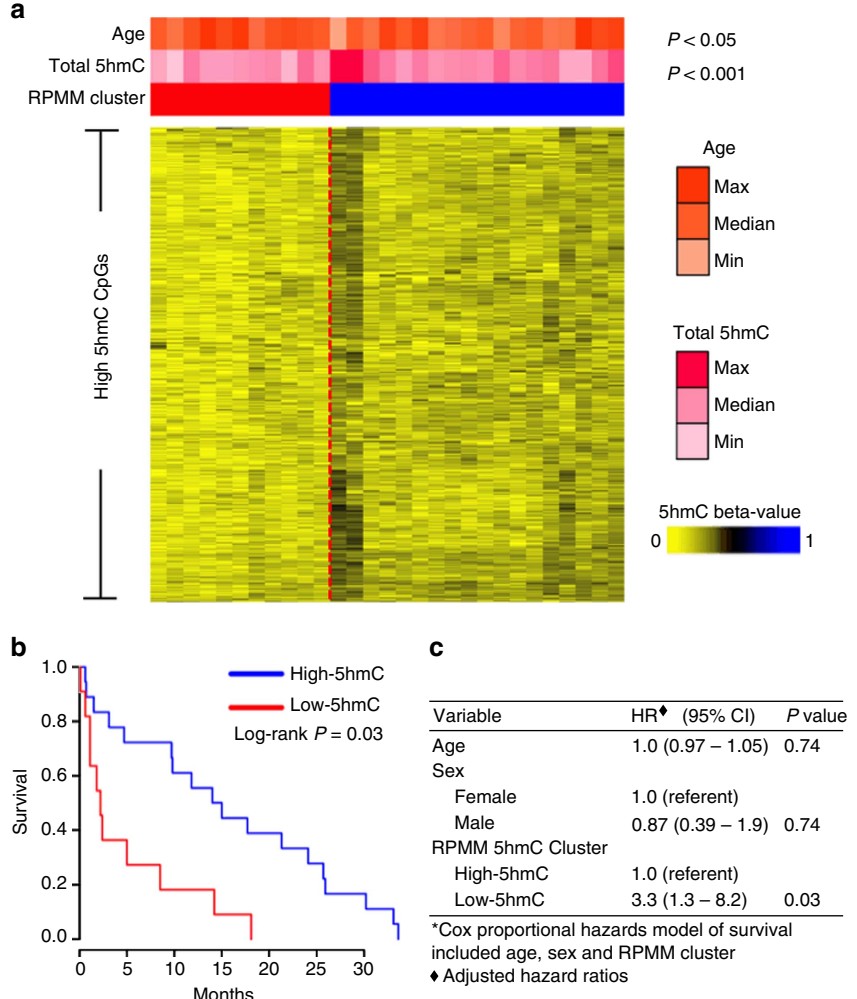

**Figure 6 | Glioblastoma 5hmC clusters are associated with survival.** (**a**) Recursively Partitioned Mixture Model (RPMM) of tumour samples based on highest 5hmC CpGs ($n = 3,876$ CpGs). In the heat map, each row represents a single CpG and each column represents a single tumour sample. Low and High 5-hydroxymethylcytosine profile clusters are indicated by colour (Low 5hmC = red, High 5hmC = blue) and are separated by a dotted red line. Total 5hmC content (quantified by the mean level of 5hmC within each tumour sample) and subject age are presented as continuous variables with intensity of colour representing the minimum and maximum values in the population of glioblastomas ($n = 30$). Cluster membership was associated with total 5hmC amount ($P = 2.0E-04$, Kruskal-Wallis rank sum test), and patient age ($P = 0.03$, Kruskal-Wallis rank sum test). (**b**) Kaplan-Meier survival plot for RPMM clusters of low 5hmC and high 5hmC. (**c**) Multivariate Cox proportional hazard ratios for survival based on RPMM cluster membership adjusted for subject age and sex.

## Discussion

In this study, we leveraged paired oxBS and BS 450 K arrays to delineate the genomic location and abundance of 5 hmC at nucleotide resolution in thirty primary glioblastomas. To the best of our knowledge, the investigation of 5 hmC genomic distribution in glioblastoma is novel. Use of 5 hmC and 5 mC profiling has permitted us to observe that the glioblastoma genome is relatively depleted of 5 hmC when compared with healthy cortex tissue, a finding consistent with studies that used less sensitive techniques[37,41–44]. We move beyond previous studies to report that despite global 5 hmC reduction[45], the genomic regions with elevated 5 hmC are strongly associated with active transcription and may be important for malignant cellular processes. Overall, our study provides evidence that a perturbed hydroxymethylome in glioblastoma may reflect progressive disruption of genomic stability and that loss of 5 hmC regulation is a potential indicator of disease progression.

Previous efforts to characterize genome-wide DNA methylation in glioblastomas using the 450 K DNA methylation arrays have demonstrated an association between an altered epigenetic state and glioblastoma tumour biology[5,8]. However, traditional BS treatment of DNA profiled on 450 K arrays alone is unable to disambiguate 5 hmC from 5 mC abundance. Existing methods to quantify the abundance of 5 hmC range from immunohistochemistry (IHC) to liquid chromatography–mass spectrometry (LC–MS) to sequence-based approaches. Both immunohistochemistry and LC–MS approaches have demonstrated that there is a depletion of 5 hmC in glioblastomas, but these techniques are unable to discern the genomic location of the 5 hmC conservation or loss[37,42–44]. In contrast, sequencing-based approaches have the ability to detect 5 hmC at the single base resolution genome-wide, but suffer from high costs that limit applicability in larger studies. To date, we are not aware of any genome-wide sequencing data for 5 hmC in glioblastoma. Notably, reduction in the complexity of the genome sequence consequent to BS modification results in increased sequencing costs[46]. The application of the oxBS and BS protocol for 450 K arrays represents an opportunity to strike a balance between genomic resolution of 5 hmC and sample throughput. Importantly, the use of paired oxBS and BS 450 K arrays has

consistently been shown to distinguish 5 hmC from both 5 mC and 5 C (refs 19,20,47).

Genome-wide loss of 5 hmC in cancer has been a widely observed, yet poorly characterized biological phenomenon[16,44]. The depletion of 5 hmC across several tumour types, including glioma, suggests that loss of 5 hmC regulation is one of the many defining features of tumorigenesis[41]. Importantly, several studies have noted that loss of 5 hmC is associated with increased expression of proliferation markers and with increasing tumour grade suggesting that loss of epigenetic regulation via 5 hmC may contribute to disease progression[37,43]. The link between decreasing 5 hmC with increasing tumour aggressiveness has been explained, in part, by the delayed generation of 5 hmC on newly synthesized DNA and the high proliferative rates of aggressive tumours[15]. Recent reports have also revealed that 5 hmC may have roles beyond transcriptional regulation. For example, 5 hmC has previously been shown to localize to DNA damage in experimental conditions and its role as an epigenetic marker of DNA damage has been shown to promote genome stability[48]. Here, we found that several of the most frequently mutated genes in glioblastoma including: *EGFR*, *PTEN*, *NF1*, *PIK3R1*, *RB1*, *PDGFRA* and *QKI* possessed high 5 hmC levels across intronic regions and further loss of 5 hmC in tumours may reflect a loss of genome integrity[13]. In contrast with prior studies, we did not observe a significant association between total 5 hmC levels and patient survival. While our method for total 5 hmC measurement was distinct from previous estimations, a lack of association between survival and global 5 hmC levels suggests that this relationship is more nuanced than previously thought. Indeed, we found that subject clusters determined by the application of RPMM to 5 hmC profiles demonstrated differences in patient survival time with the low-5 hmC tumours having a significantly worst prognosis independent of subject age. Together, our analysis provides a more complete understanding of dynamic cytosine modifications that may contribute to disease phenotypes and genomic instability.

Not considered by previous studies is the possibility that cancer cell conservation of 5 hmC may also be critical for glioblastoma-genesis and tumour growth as suggested by Takai *et al.*[17]. Indeed, the highest levels of 5 hmC found in this cohort of glioblastomas were in regions of low CpG-content, a phenomenon also present in acute myeloid leukaemia and non-diseased tissue[22,26,49,50]. Genes with high 5 hmC were noted to be located within actively transcribed genes in glioblastoma highlighting the role of 5 hmC as a potential facilitator of transcription. 5 hmC was also preferentially found in glioblastoma-specific enhancers and super-enhancers suggesting that deposition of 5 hmC at these gene regulatory regions may indicate epigenetic states permissive to cancer cell survival. Prior work has demonstrated that 5 hmC modulates enhancer activity and regulates gene expression programs during cellular differentiation suggesting that 5 hmC deregulation may impact the dedifferentiation observed in glioblastoma[50–53]. The enrichment of cellular identity pathways in our GREAT analysis and the binding sites of transcription factors that drive cellular proliferation further supported the link between 5 hmC and disease-critical gene expression programs. Indeed, the enrichment of immune response and somatic stem cell maintenance may reflect cellular subpopulations (that is, glioblastoma stem cells and infiltrating immune cells) necessary for continued tumour growth[54]. Across this cohort, the consistent genomic localization of 5 hmC suggests a role for 5 hmC in genomic regulation of glioblastoma cells that merits future investigation.

In summary, we have generated nucleotide resolution maps of 5 hmC in thirty glioblastomas and linked 5 hmC with gene regulatory regions. Our results demonstrate that the glioblastoma genome exhibits a global loss of 5 hmC compared with healthy prefrontal cortex tissues, but observed regions of conserved 5 hmC implying novel associations between 5 hmC and critical tumour transcriptional programs. Our study also highlights that 5 hmC may, in a manner similar to 5 mC, serve as marker of cellular identity and that genomic patterns reflect cellular states in tissue samples. Thus, methods presented here that defined associations between 5 hmC distribution and genomic features may be more broadly applicable to other diseases as well as characterization of 5 hmC function in non-diseased tissue.

## Methods

**Study population.** Pathologically confirmed fresh frozen glioblastoma specimens from 30 subjects diagnosed at Dartmouth Hitchcock Medical Center (Lebanon, NH, USA) between 2004 and 2012 were identified for study. All subjects provided written informed consent at the time of surgery for use of their tumour specimens in research as approved by the committee for protection of human subjects (Institutional review board). Subject demographic, tumour characteristic and survival follow-up were available for study and all tissues accessed were from deceased subjects. This work was performed in accordance with the ethical principles in the Declaration of Helsinki.

**DNA (hydroxy)methylation microarray profiling.** Tumour DNA was extracted with the QIAmp DNeasy tissue kit (Qiagen) according to the manufacturer's instructions. DNA quantity and quality was assessed with the Qubit 3.0 fluorometer (Life Technologies). Sample DNA was subjected to tandem BS and oxidative BS (oxBS) conversion with an input of 4 μg per sample using the TrueMethyl kit v.1.1 (Cambridge Epigenetix) protocol optimized for Illumina HumanMethylation450 arrays. Prior publications have validated 5 hmC values in human brain tissue by oxBS-independent approaches and have demonstrated that replicate samples treated with BS–oxBS display a high level of reproducibility[19]. Following quantification genomic DNA was sheared to ~10 kb fragments using g-TUBE (Covaris), and purified with the Gene-JET PCR Purification kit (Thermo Scientific). A total of1.4 μg was carried forward through oxBS conversion with the TrueMethyl protocol, and 1.05 μg through the BS conversion arm of the protocol with manufacturer recommended mass and volume. Recovered substrate ssDNA was quantified with Qubit and submitted for DNA methylation array processing at the UCSF genomics core facility.

***IDH1* and *IDH2* mutation.** All glioblastomas were sequenced for *IDH1* (R132) and *IDH2* (R140 and R172) mutation status using PCR amplification and Sanger sequencing. Briefly, 10 ng of genomic DNA was amplified with primers for each of *IDH1* spanning codon 132: F-GGTGGCACGGTCTTCAGAG, R-ATGTGTTGA GATGGACGCCT, and *IDH2* spanning codons 140 and 172: F-TTCTGGTTGA AAGATGGCG, and R-GGATGGCTAGGCGAGGAG. Reaction conditions included a denaturation at 94 °C for 2 min followed by 35 cycles of 94 °C for 30 s, 62 °C for 30 s and 68 °C for 1 min with extension at 68 °C for 5 min as previously published[55].

**Data processing and statistical analysis.** All data analysis was conducted in R version 3.1.2. Normalization and background correction of raw signals from each of the BS and oxBS converted samples was achieved using the *FunNorm* procedure available in the R/Bioconductor package *minfi* (version 1.10.2)[56]. Before analysis we removed CpG sites on sex chromosomes as well as those corresponding to probes previously identified as cross-reactive or containing SNPs[57], resulting in 387,617 CpGs remaining for analysis. We applied a novel technique for estimating 5 mC, 5 hmC and unmethylated proportions. Briefly, each CpG corresponded to a data vector ($S_{BS}$, $R_{BS}$, $S_{oxBS}$, $R_{oxBS}$), with $R_k$ representing total signal (unmethylated + methylated) and representing methylated signal ($k \in \{BS, oxBS\}$); we used maximum likelihood to fit the data generating model:

$$S_{BS} \sim \text{Beta}(R_{BS}(\pi_2 + \pi_3), R_{BS}\pi_1), \ S_{oxBS} \sim \text{Beta}(R_{oxBS}\pi_2, R_{oxBS}(\pi_1 + \pi_3)) \quad (1)$$

under the constraints $\pi_j > 0$ ($j \in \{1, 2, 3\}$), $\pi_1 + \pi_2 + \pi_3 = 1$

The resulting estimates parameters (unmethylated proportion), (5 mC proportion) and (5 hmC proportion). Note that this method explicitly disallows negative proportions, although we did observe numerically zero values of 5 hmC ($\pi_3 < 10^{-16}$). R code for maximum likelihood estimation as applied to 450 K arrays is available in the R-package 'OxyBS'. Since the raw intensity data (IDAT) files were unavailable for the prefrontal cortex 5 hmC data set (GSE74368) we processed the samples using the alternative naïve subtraction method outlined in Houseman *et al.*[21]. The global level of 5 hmC for each sample in both the glioblastoma and prefrontal cortex samples was determined by summing the 5 hmC beta-values for all CpGs within in each sample and dividing by the total number of CpGs that passed quality control metrics and were considered in our analyses ($n = 387,617$).

For subsequent analysis, we used array annotation available in the R/Bioconductor packages IlluminaHumanMethylation450kmanifest, version 0.4.0

and IlluminaHumanMethylation450kanno.ilmn12.hg19, version 0.2.1. In particular, we classified each CpG by genomic context (CpG island, shore, shelf or ocean)[58].

We moved to delineate the level at which we could confidently call a genomic location as possessing *bona fide* 5 hmC. In our analysis, we defined the locations within glioblastoma that have the 1% highest mean 5 hmC level to be considered high 5 hmC CpGs. Next, each CpG was assigned a classifier for high 5 hmC (that is, 0 or 1), putative enhancer regions, open chromatin, Polycomb group target genes, and gene regions (that is, four separate vectors for promoter, exons, introns or intergenic) as well as the CpG's relationship to a CpG island (that is CpG island, shore, shelf and ocean). Typically, a Fisher's test is used to test for enrichment of these features; however, we decided to use the Cochran–Mantel–Haenszel test as this approach permits stratification by CpG island probe type, which we demonstrated has a substantial impact of the level of 5 hmC.

**RNA extraction and gene expression.** Tumour RNA was extracted with the RNeasy Mini tissue kit (Qiagen) according to the manufacturer's instructions. Insufficient RNA was available for gene expression from six tumours (Supplementary Data 4). RNA quantity and quality was assessed with the Qubit 3.0 fluorometer (Life Technologies). The nCounter Analysis System (NanoString Technologies) was used to simultaneously assess the absolute expression of 41 genes per subject (n = 24). Candidate gene transcripts of interest are presented in Supplementary Data 4 and were included because one of the following selection criteria: (i) epigenetic enzymes (that is, *DNMT1*),[48] genes with a high proportion of CpGs with high 5 hmC in a given gene region with known multiple transcription splice variants (that is, *RGMA* 5′UTR), (iii) genes with a known relation to glioblastoma pathobiology (that is, *MGMT*) and (iv) glioblastoma housekeeping genes. The digital multiplexed NanoString nCounter for custom gene set was performed according to manufacturers' instruction with total RNA. Data normalization was performed using the nSolver Analysis software (NanoString, V2.6) with initial positive controls used to normalize all platform associated sources of variation and reference gene normalization was performed using housekeeping genes (*PUM1*, *GUSB*, *TBP*, *ACTB* and *SDHA*).

**Genomic region enrichment analysis.** To examine whether high 5 hmC CpGs were associated with specific gene sets we used the GREAT software and to query whether 5 hmC was associated with TFBSs in ENCODE we used the LOLA software[33,59]. In both of these analyses, our query input set of genomic regions to be tested for enrichment were the genomic locations of the high 5 hmC CpGs and the background set were the genomic locations of the 387,617 CpGs used in the analysis. The reference database for the GREAT analysis was selected under the default setting and the ENCODE TFBSs that included 689 different ChIP-seq experiments was selected for the LOLA analysis.

**Estimation of cellular proportions using DNA methylation data.** The RefFreeEWAS algorithm (R-package RefFreeEWAS) is a method for the reference-free deconvolution that provides proportions of putative cell types as defined by their underlying methylomes[60]. Previously, this algorithm has been shown to reasonably estimate the number of constituent cell types[61]. Moreover, RefFreeEWAS is a variant of non-negative matrix factorization and is similar to approaches that use gene expression levels to estimate the proportion of normal tissue cells in a tumour sample. Briefly, we used the 10,000 most variable CpGs in terms of their DNA methylation values across all samples (that is, oxBS estimates alone) and identified the optimal number of cell types to be three. We then used the RefFreeCellMix function across all 387,617 CpGs to define sample-specific estimates of cellular proportions. Notably, the glioblastoma samples demonstrated significant proportions of two distinct putative cell types (Supplementary Fig. 3A). We chose the putative cell-type that explained the greatest variation in cellular proportions to present estimates of tumour purity.

**Copy number alteration calls from Illumina 450 K arrays.** To determine common copy number alterations in primary glioblastomas including: *EGFR* gain, *CDKN2A* loss, chromosome 7 gain and chromosome 10 loss, we used the Bioconductor package *CopyNumber450 K* and confirmed copy number gain in samples where *EGFR* expression was available.

**Definition of genomic regions.** We defined regions of the genome (for example, introns, exons, promoters) using the UCSC_hg19_refGene file from the UCSC Genome Browser and collapsed 5 hmC and 5 mC around the promoter region using the genomation bioconductor package[62].

**TCGA glioblastoma gene expression.** Level 3 normalized RNAseqV2 expression levels from TCGA were binned into tertiles based upon mean expression across all samples. Glioblastoma mRNA splicing data was downloaded from the TCGASpliceSeq database (that is, mRNA splicing patterns of TCGA RNAseq data). The data was downloaded from the database with the following filters: TCGA disease type was set as glioblastoma, all genes were considered, all TCGA glioblastoma samples were considered (n = 160 for splicing), all splice events

(that is, exon skip, alternate promoters and so on), and software specific filters of Percent Splice In value (PSI = number of transcript element reads amid all RNA sequencing reads covering the splicing event) of 75% and a minimum range of 10 PSI values to identify variable splicing events across tumours.

**Code availability.** The R code for the OxyBS algorithm can be found in Houseman *et al.*[21].

**Data availability.** The glioblastoma 5 hmC and 5 mC DNA microarray data that support the findings of this study have been deposited in the Gene Expression Omnibus (GEO) with the accession code GSE73895 (http://www.ncbi.nlm.nih.gov/geo/). Five prefrontal cortex samples from individuals with no evidence of neurological impairment as described in Lunnon *et al.*[28] were accessed on GEO (GSE74368). Glioblastoma-specific enhancer and super-enhancer coordinates were obtained from U87 cells as described in Hnisz *et al.* (GSE51522)[30]. Enhancer-associated H3K27ac ChIP-seq coordinates from three freshly resected primary glioblastoma were processed as described in Suva *et al.* (GSE54792)[63]. DNase hypersensitivity sites in H54 glioblastoma cells (ENCODE) were used to determine potential areas of open chromatin specific to the disease (GSM816668). Level 1 TCGA glioblastoma (GBM) 450 K DNA methylation data and associated clinical data were obtained from the TCGA data portal (http://cancergenome.nih.gov/) and processed using the *minfi* Bioconductor package[56]. GBM level 3 gene expression data from the RNAseqV2 platform were also obtained from the TCGA data portal (http://cancergenome.nih.gov/). All remaining data is available within the article, Supplementary Information files or from the authors upon request.

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

## Acknowledgements

We kindly acknowledge our colleague Owen M. Wilkins for suggestions and discussions that improved the manuscript. This work was supported by the National Institutes of Health; grant numbers R01 DE022772 to B.C.C., R01 MH094609 to E.A.H. The research reported in this publication was also supported by the Center for Molecular Epidemiology COBRE program with grant funds from the National Institute of General Medical Sciences (NIGMS) of the National Institutes of Health (NIH) under award number P20 GM104416.

## Author contributions

K.C.J. conceived and designed the approach, carried out laboratory experiments, performed statistical analyses, interpreted the results, wrote and revised the manuscript. E.A.H. conceived and designed the approach, generated statistical models, performed statistical analyses, interpreted the results, wrote and revised the manuscript. J.E.K. carried out laboratory experiments and revised the manuscript. K.M.v.H. carried out laboratory experiments and revised the manuscript. C.E.F. conceived and designed the approach, interpreted the results, and revised the manuscript. B.C.C. conceived and designed the approach, interpreted the results, wrote and revised the manuscript. All authors have read and approved the final manuscript.

## Additional information

**Competing financial interests:** The authors declare no competing financial interests.

