## [Peer Review File · Nature Communications]

Reviewer #1 (Remarks to the Author)

The authors examine genome-wide profiles of 5-methylcytosine and 5-hydroxymethylcytosine in glioblastoma. They show that 5-hydroxymethylcytosine is depleted in glioblastoma compared with prefrontal cortex tissue. They correlate genomic localization of 5-hydroxymethylcytosine in glioblastoma with gene function, including binding sites of transcription factors that drive cellular proliferation. Low 5-hydroxymethylcytosine patterns were correlated with poorer overall survival.

1. The validity of using the pre published data on prefrontal cortex as 'normal control' needs to be addressed. Epigenetic phenomena are cell type specific, and prefrontal cortex likely represents multiple cell types combined, and may not be representative of normal cell 5hmC as a comparison.
2. The correlation of 5hmC to the list of 'genes important in GBM' is not well justified. The functional significance of what 5hmC is doing related to gene expression is not clearly shown. 5hmC has been associated with gene expression and silencing events.
3. There is correlative data, but a lack of any functional data to suggest role/effect of 5hmC at GBM super enhancers. Experimental evidence in this regard would solidify this connection.
4. The survival correlation is quite visually impressive, but I am concerned whether there is overfitting and that there is a lack of validation, which is always a concern when a model is fitted to patient outcome in a small sample set. It is concerning that one of the methods used did not show a survival correlation whereas another one did. It is also unclear whether the findings related to survival are unique to the 3876 probes used. Patients whose tumors were in the low methylation group (with the worse outcome) were also much older than in the high methylation group. Since older age is the most important and reproducible factor related to poorer prognosis in GBM, this raises the question as to whether methylation status is a surrogate for patient age. Validation of these data, for example in an independent set of samples, would be most helpful to bolster this survival association.
5. Given the fundamental importance of IDH mutation on the glioma epigenome (as the authors have noted), the lack of IDH-mutated samples stands out as striking and as a gap in this study. What is the difference, if any between hydroxymethylation patterns in IDH-mutated versus -wild type glioma?
6. It is unclear why EGFR amplification is singled out by the authors as a relevant marker for comparative purposes, relative to other known markers/aberrations (TP53, PTEN, TERT promoter, CDKN2A, etc).

Reviewer #2 (Remarks to the Author)

The manuscript entitled "5-Hydroxymethylcytosine localizes to glioblastoma-specific enhancer elements that stratifies patient survival" investigated abnormalities of 5hmC in glioblastoma primary tumors by examining the localization pattern of 5hmC as well as potential association with gene regulation and clinical outcomes. This study revealed possible novel functional relevance of 5hmC to glioblastoma etiology and survival rate using data mining as a major tool. However, the study is rather observational and lacking in depth of analysis as many conclusions are established at association level while the overall mechanistic support is weak. The manuscript started with comparison between glioblastoma and normal cortex tissue, but never performed any comparative analysis in the following results. It mainly focused within glioblastoma samples, which yielded the potential interesting point of the tumor-specific pattern of 5hmC. The analysis results observed in this study centered at glioblastoma tissue may therefore hold true for normal tissue as well. Some suggestions/questions/issues to be addressed are as follows:

1. Figure 1 is not revealing any part of the results and therefore should be a supplemental figure.
2. The method for measuring global level of 5hmC was not stated in the method. Statistical difference of 5hmC between prefrontal cortex and GBM should be evaluated.
3. Although authors hypothesized that tumor levels of 5hmC may be reflective of TET enzymatic activity, this was not eventually tested in the manuscript, and several studies have already shown that 5hmC and TET levels do not correlate.
4. In figure 2D, how the quantiles were decided was not clearly described in the results nor the

figure legends. Statistical assessment across island strata should be shown for both 5mC and 5hmC.

5. The sentence "5hmC...shown to localize to DNA damage in tumor cells...." In the result section "Elevated levels of 5hmC localize to key glioblastoma genes" need to be clarified.

6. In figure 3A, what genomic feature are 5hmC localized to in those most frequently mutated genes? The impact of this observation should be discussed in the discussion section with reference support. Was figure 3B plotted with all probes or just the selected highest CpGs? Figure 3A and 3B doesn't go along in the same result section as they are hitting different points.

7. Figure 3C was not clearly explained in the results.

8. Since the enhancer/super-enhancer mapping was not performed with any patients' samples used in this study, at least a few loci should be confirmed by ChIP-qPCR as cell lines could differ greatly from tissue samples. The authors also need to clarify the cell type origin of enhancer indexed by the 450K annotation.

9. There should be a statistical assessment for the association between 5hmC and gene expression in figure 5.

10. Supplemental figure 6A: TCGA G-CIMP should be shown alongside the patient samples investigated in this study for the purpose of comparison and excluding true G-CIMP+ sample.

Response to reviewer's comments on the manuscript: "5-Hydroxymethylcytosine localizes to glioblastoma-specific enhancer elements that stratifies patient survival" (ORCID ID: 0000-0002-3385-0322)

Reviewer #1 (Remarks to the Author):

The authors examine genome-wide profiles of 5-methylcytosine and 5-hydroxymethylcytosine in glioblastoma. They show that 5-hydroxymethylcytosine is depleted in glioblastoma compared with prefrontal cortex tissue. They correlate genomic localization of 5-hydroxymethylcytosine in glioblastoma with gene function, including binding sites of transcription factors that drive cellular proliferation. Low 5-hydroxymethylcytosine patterns were correlated with poorer overall survival.

1. The validity of using the pre published data on prefrontal cortex as 'normal control' needs to be addressed. Epigenetic phenomena are cell type specific, and prefrontal cortex likely represents multiple cell types combined, and may not be representative of normal cell 5hmC as a comparison.

RESPONSE: The reviewer raises an excellent point. The primary objective of the discussed analysis was to evaluate the consistency of our novel 5hmC measurement with previous studies that have demonstrated lower levels of 5hmC, via immunohistochemistry measurements, in glioma versus normal brain^{1,2}. Our decision to use prefrontal cortex as a comparative normal brain control was informed and is substantiated by previous studies that have used this referent normal tissue for the identification of differentially methylated regions in glioblastoma^{3,4}. Nevertheless, we agree with the reviewer that the prefrontal cortex likely represents a mixture of both neuronal and glial cells for which distinct DNA methylation patterns have been observed⁵. We now explicitly acknowledge that mixed cellular composition of prefrontal cortex is a potential limitation for this comparison with text on p. 4.

2. The correlation of 5hmC to the list of 'genes important in GBM' is not well justified. The functional significance of what 5hmC is doing related to gene expression is not clearly shown. 5hmC has been associated with gene expression and silencing events.

RESPONSE: The statement regarding 'genes important in GBM' was more ambiguous than originally intended and has been moved from the Results to Discussion section with the revisions seen below (p. 9):

"Recent reports have also revealed that 5hmC may have roles beyond transcriptional regulation. For example, 5hmC has previously been shown to localize to DNA damage in experimental conditions and its role as an epigenetic marker of DNA damage has been shown to promote genome stability⁶. Here, we found that several of the most frequently mutated genes in glioblastoma

including: *EGFR*, *PTEN*, *NF1*, *PIK3R1*, *RB1*, *PDGFRA*, and *QKI*⁷ possessed high 5hmC levels across intronic regions and further loss of 5hmC in tumors may reflect a loss of genome integrity.”

In response to this reviewer’s second concern, we have revised the presentation of our analyses comparing 5hmC levels with gene expression. Consistent evidence between studies in non-tumor tissue⁸⁻¹⁰, and our data indicate that increased 5hmC is correlated with higher gene expression levels. More specifically, in our data high 5hmC CpGs tracked to genes actively transcribed in glioblastoma (i.e. Supplementary Table 5), and increasing 5hmC was generally positively correlated with gene expression in glioblastomas (i.e. Figure 4A). These results for 5hmC are in contrast with well-established negative correlations between gene expression and 5mC (Figure 4B). Of course, there will invariably be exceptions to the generalized observation that 5hmC is associated with increased expression and determination of causality at distinct gene regions requires additional investigation. Our findings extend prior observations of elevated 5hmC associated with increased expression in non-tumor tissues to tumor tissue^{8,11}, and provide nucleotide-resolution data that will allow comparison of 5hmC levels with gene expression among other tumor types in future work. To this end, we have added further discussion to the interpretation our gene expression results on p. 10.

3. There is correlative data, but a lack of any functional data to suggest role/effect of 5hmC at GBM super enhancers. Experimental evidence in this regard would solidify this connection.

RESPONSE: This is a valid concern. We agree that additional experiments designed with stimuli and high-resolution technologies are needed to understand the underlying mechanisms of 5hmC function at super-enhancers. Indeed, some early studies of aberrant super enhancer DNA methylation in cancer have sought to elucidate functional consequences by measuring expression of super-enhancer associated genes¹². However, while we share the reviewer’s interest in elucidating the functional consequences of 5mC and 5hmC at enhancer elements, we believe such experiments are beyond the aims of this manuscript. Nonetheless, we have conducted additional analyses to confirm that 5hmC CpGs localize to enhancer elements in experiments derived from primary human tumors¹³. Specifically, genomic regions of high 5hmC were significantly enriched at enhancer elements from primary glioblastomas investigated in Suva et al¹³ (MGH27 tumor OR = 1.6, $P = 1.4E-28$; MGH28 OR = 3.0, $P = 1.9E-211$; and MGH30 OR = 3.1, $P = 3.2E-211$) suggesting that enrichment of enhancer elements *in vitro* are also present in human tumors. Overall, we believe that the observed enrichment of 5hmC to enhancers in glioblastoma cell lines and primary tumors are discoveries that provide a foundation for future in-depth functional characterization.

4. The survival correlation is quite visually impressive, but I am concerned whether there is overfitting and that there is a lack of validation, which is always a concern when a model is fitted to patient outcome in a small sample set. It is concerning that one of the methods used did not show a survival correlation whereas another one did. It is also unclear whether the findings related to survival are unique to the 3876 probes used. Patients whose tumors were in the low methylation group (with the worse outcome) were also much older than in the high methylation group. Since older age is the most important and reproducible factor related to poorer prognosis in GBM, this raises the question as to whether methylation status is a surrogate for patient age. Validation of these data, for example in an independent set of samples, would be most helpful to bolster this survival association.

RESPONSE: The reviewer's concerns are understandable and we have clarified our rationale for using two separate approaches to test the relation of 5hmC levels with survival below and in the manuscript. First, we sought to compare survival of subjects using an overall summary measure of 5hmC as other groups in prior work have measured summary 5hmC levels with different approaches (e.g. IHC, LC-MS). One of these prior investigations in glioblastoma had observed that lower total 5hmC content measured by immunohistochemistry was related with poorer survival in univariate analysis and multivariable models adjusted for age and sex ($n = 52$ adult glioblastomas)¹⁴. In our study, we did not observe a significant association between survival and the measure of total 5hmC. Next, to take advantage of our CpG-specific data without employing feature-by-feature statistical inference tests that could introduce type I error, we used a clustering approach to define classes of samples based on CpG-specific 5hmC levels for CpGs with the highest levels of 5hmC. It should be noted that this classification-based approach is not expected to perform similarly to the "total 5hmC" method because the clustering approach borrows statistical strength across multiple context-dependent CpG associations while the "total 5hmC" measure assesses a single unidirectional association test. Still, the clustering approach does have a drawback of being dependent on the number of loci used in the clustering as the reviewer has suggested. To this end, we also performed separate clustering analyses of the 1000, 2000, 3000, and 4000 highest 5hmC probes and observed complete stability of cluster membership (i.e., samples remained in either low or high 5hmC clusters regardless of probe number selected). Finally, the reviewer has expressed concerned over whether 5-hydroxymethylation status is a surrogate for subject age as older age is recognized to be associated with poorer prognoses. In the original manuscript, to investigate the relation of 5hmC clusters with survival we plotted the Kaplan-Meier strata for 5hmC clusters and used both a Log-rank test and fit a Cox proportional hazards model adjusted for age and sex. The Cox model demonstrated that patients in the 5hmC cluster with low 5hmC levels (relative to the other cluster) had significantly reduced survival independent of age and patient sex (HR = 3.3, 95% CI 1.3 - 8.2, $P = 0.03$). Taken together, these

exploratory survival analyses revealed a robust association within our cohort and future studies will be needed to confirm the existence of these patterns beyond the present study.

5. Given the fundamental importance of IDH mutation on the glioma epigenome (as the authors have noted), the lack of IDH-mutated samples stands out as striking and as a gap in this study. What is the difference, if any between hydroxymethylation patterns in IDH-mutated versus -wild type glioma?

RESPONSE: It is true that IDH mutation is fundamentally important in glioma. However, primary glioblastoma is a high-grade glioma where only ~5-10% of tumors have an IDH mutation, whereas ~80% of low-grade gliomas (astrocytomas, oligodendrogliomas, and others), have an IDH mutation^{15,16}. Studies of 5hmC and 5mC patterns in low-grade gliomas comparing IDH-mutant and wild type tumors should be a priority for future studies.

6. It is unclear why EGFR amplification is singled out by the authors as a relevant marker for comparative purposes, relative to other known markers/aberrations (TP53, PTEN, TERT promoter, CDKN2A, etc).

RESPONSE: It is possible to use high-dimensional DNA methylation arrays to profile copy number alterations¹⁷ and, accordingly, we chose to assess *EGFR* status for its known role in glioblastoma pathobiology. In particular, *EGFR* amplification status was included as a relevant comparative marker because it is one of the most common oncogenic events in glioblastoma^{18,19}, and has been shown to repress the TET enzymes²⁰. We acknowledge that a copy number alteration analysis need not be limited to *EGFR* amplification alone. We have added other common copy number alterations in glioblastomas²¹ and added Supplemental Figure 7: *CDKN2A* loss, gain of chromosome 7, and deletion of chromosome 10. Notably, we did not observe enrichment for any of the above copy number alterations with 5hmC cluster membership. Future work will be needed to evaluate the potential relation of 5hmC patterns with the status of other recurrently mutated genes in glioblastoma (i.e., *PTEN*, *TP53*, and *TERT* promoter).

Reviewer #2 (Remarks to the Author):

The manuscript entitled "5-Hydroxymethylcytosine localizes to glioblastoma-specific enhancer elements that stratifies patient survival" investigated abnormalities of 5hmC in glioblastoma primary tumors by examining the localization pattern of 5hmC as well as potential association with gene regulation and clinical outcomes. This study revealed possible novel functional relevance of 5hmC to glioblastoma etiology and survival rate using data mining as a major tool. However, the study is rather observational and lacking in depth of analysis as many conclusions are established at association level while the overall

mechanistic support is weak. The manuscript started with comparison between glioblastoma and normal cortex tissue, but never performed any comparative analysis in the following results. It mainly focused within glioblastoma samples, which yielded the potential interesting point of the tumor-specific pattern of 5hmC. The analysis results observed in this study centered at glioblastoma tissue may therefore hold true for normal tissue as well. Some suggestions/questions/issues to be addressed are as follows:

1. Figure 1 is not revealing any part of the results and therefore should be a supplemental figure.

RESPONSE: As requested, Figure 1 is now Supplemental Figure 1.

2. The method for measuring global level of 5hmC was not stated in the method. Statistical difference of 5hmC between prefrontal cortex and GBM should be evaluated.

RESPONSE: We regret this omission and now include additional text in the Methods (p. 13) and Results sections (p. 4) describing the construction and application of global 5hmC. Briefly, the global level of 5hmC for each sample was determined by summing the 5hmC beta-values for all CpGs within in each sample and dividing by the total number of CpGs that passed QC metrics and were considered in our analyses ($n = 387,617$). This global measure was constructed to allow comparison with previous studies that used other methods to measure total genomic 5hmC and to provide a simple summary measure, represented as a percentage of 5hmC content from the oxBS 450K array. We have added a comparison of global 5hmC levels between GBM and normal prefrontal cortex to the results section (p. 4). We observed a 3.5 fold decrease in 5hmC content in glioblastoma compared with prefrontal cortex (Wilcoxon rank sum test, $P = 6.2E-06$).

3. Although authors hypothesized that tumor levels of 5hmC may be reflective of TET enzymatic activity, this was not eventually tested in the manuscript, and several studies have already shown that 5hmC and TET levels do not correlate.

RESPONSE: We agree that our explanation for 5hmC levels reflecting TET enzymatic activity is speculative and have removed this language from the Results section. In addition, we added text on p. 4 to support that 5hmC levels were not correlated with TET gene expression levels as observed in previous studies¹⁰.

4. In figure 2D, how the quantiles were decided was not clearly described in the results nor the figure legends. Statistical assessment across island strata should be shown for both 5mC and 5hmC.

RESPONSE: We thank the reviewer for bringing this omission to our attention. The percentiles presented in the original Figure panels 2C and 2D are

descriptions of quantiles relative to 100. These quantiles/percentiles were selected arbitrarily to demonstrate that levels of 5mC/5hmC vary based on CpG island context. The selection of quantiles has now been clarified in the Figure legend and in the Figure we now indicate statistical significance for comparisons of the differences in beta-values at each quantile across CpG island strata from Kruskal-Wallis tests. Statistical significance at a Bonferroni adjusted alpha ($P = 8.3E-03$) has been indicated by an asterisk (“**”) at each strata in revised Figure 1C and 1D.

5. The sentence "5hmC...shown to localize to DNA damage in tumor cells...." In the result section "Elevated levels of 5hmC localize to key glioblastoma genes" need to be clarified.

RESPONSE: The statement regarding “5hmC...shown to localize to DNA damage in tumor cells...” was meant to provide context for our findings and was more ambiguous than intended. The sentence refers to previous work that demonstrated 5hmC is actively enriched at DNA damage sites in cancer cell lines and that experimental induction of DNA damage increases 5hmC levels ⁶. We have moved this text to the Discussion section and added clarification:

“Recent reports have also revealed that 5hmC may have roles beyond transcriptional regulation. For example, 5hmC has previously been shown to localize to DNA damage in experimental conditions and its role as an epigenetic marker of DNA damage has been shown to promote genome stability. Here, we found that several of the most frequently mutated genes in glioblastoma including: *EGFR*, *PTEN*, *NF1*, *PIK3R1*, *RB1*, *PDGFRA*, and *QKI* possessed high 5hmC levels across intronic regions and further loss of 5hmC in tumors may reflect a loss of genome integrity.”

Furthermore, we agree that the Results section title “Elevated levels of 5hmC localize to key glioblastoma genes” lacked clarity and we have revised this section header to “Elevated levels of 5hmC are uniquely distributed in the glioblastoma genome and enriched for gene regulatory regions.”

6. In figure 3A, what genomic feature are 5hmC localized to in those most frequently mutated genes? The impact of this observation should be discussed in the discussion section with reference support. Was figure 3B plotted with all probes or just the selected highest CpGs? Figure 3A and 3B doesn't go along in the same result section as they are hitting different points.

RESPONSE: For Figure 3A, the genomic features to which 5hmC localizes are as follows: *EGFR* (gene body, intron), *PTEN* (gene body, intron), *NF1* (gene body, intron), *PIK3R1* (TSS200, promoter), *RB1* (gene body, intron), *PDGFRA* (gene body, exon), and *QKI* (gene body, intron). Interestingly, the levels of 5hmC at the most frequently mutated genes were most abundant at the intronic regions, regions that may not have been sequenced in tumor samples profiled by whole

exome sequencing. The following text has been added to the Discussion section:

“Recent reports have also revealed that 5hmC may have roles beyond transcriptional regulation. For example, 5hmC has previously been shown to localize to DNA damage in experimental conditions and its role as an epigenetic marker of DNA damage has been shown to promote genome stability⁶. Here, we found that several of the most frequently mutated genes in glioblastoma including: *EGFR*, *PTEN*, *NF1*, *PIK3R1*, *RB1*, *PDGFRA*, and *QKI*⁷ possessed high 5hmC levels across intronic regions and further loss of 5hmC in tumors may reflect a loss of genome integrity.”

Figure 3B was plotted with the n = 3876 selected highest 5hmC CpGs and this is now included in the figure legend. We thank the reviewer for pointing out the discontinuity between Figures 3A and 3B. Figure 3A has been made a supplementary figure (Supplementary Figure 5), which should improve the readability of the figure.

7. Figure 3C was not clearly explained in the results.

RESPONSE: We thank the reviewers for bringing this to our attention. We have added language to the figure legend and Results section of the revised manuscript (Figure 3C is now Figure 2B), as follows:

“Figure 2B shows the distribution of high 5hmC CpG sites relative to the nearest canonical transcriptional start site in base pairs²² with category bins for genomic distance both upstream and downstream of the TSS.”

We have also added the numeric proportions of high 5hmC CpGs at each genomic distance to TSS category.

8. Since the enhancer/super-enhancer mapping was not performed with any patients' samples used in this study, at least a few loci should be confirmed by ChIP-qPCR as cell lines could differ greatly from tissue samples. The authors also need to clarify the cell type origin of enhancer indexed by the 450K annotation.

RESPONSE: We thank the reviewer for this constructive comment. We have added analyses of enhancer regions as assessed in several freshly resected primary glioblastomas¹³. Chromatin immunoprecipitation followed by next-generation sequencing (or qPCR) requires ample substrate and poses technical challenges²³ that are not possible to overcome with the limited archival tissue substrate in the same samples for which we measured 5hmC/5mC. In its place, we have analyzed enhancer ChIP-seq data from the three primary glioblastomas presented in Suva *et al Cell* 2014¹³. We have added to our manuscript analyses that confirm enrichment for enhancer elements among high 5hmC CpGs using enhancer regions defined in primary human glioblastomas. Specifically, genomic

regions of high 5hmC were significantly enriched in enhancer elements of all three primary glioblastomas (MGH27 tumor OR = 1.6, $P = 1.4E-28$; MGH28 tumor OR = 3.0, $P = 1.9E-211$; and MGH30 tumor OR = 3.1, $P = 3.2E-211$) suggesting that enrichment of enhancer elements we observed previously is not unique to enhancers defined in cell lines. Additionally, in the prior version of the manuscript we used the 450K informatically-identified enhancers from Illumina²⁴ which define enhancer locations by leveraging multiple ENCODE cell-line data sets. While this approach is valid, a more cell-type specific definition of enhancers is achievable by analyzing the cell type of interest. To this end, we turned to glioblastoma-specific enhancer data from the U87 glioblastoma cell line²⁵ and observed a statistically significant enrichment of high 5hmC CpGs at glioblastoma cell-line defined enhancers (OR = 2.2, $P = 1.7E-46$), this analysis is now presented on p. 6 of the Results section. We believe the addition of these new analyses based on the suggestion of the reviewer has strengthened our findings.

9. There should be a statistical assessment for the association between 5hmC and gene expression in figure 5.

RESPONSE: We regret that our statistical assessment in Figure 5 was unclear. The size of each bubble represents the $-\log_{10}(P\text{-value})$ and in the previous version of the manuscript the legend was provided for Figure 5B, but not Figure 5A. This has been corrected in the revised manuscript. Further, the data for Figure 5 is presented in tabular form in Supplementary Table 7.

10. Supplemental figure 6A: TCGA G-CIMP should be shown alongside the patient samples investigated in this study for the purpose of comparison and excluding true G-CIMP+ sample.

RESPONSE: We thank the reviewer for their constructive comment as such a comparison would confirm the G-CIMP+ sample in the present cohort. We now show that the G-CIMP+ in our cohort (panel A, far left sample) resembles the DNA methylation profiles of G-CIMP+ samples (panel B, far left) in the TCGA 450K glioblastoma data set²⁶. Supplemental Figure 6A has now been revised to include both the data from our samples as well as the TCGA (see legend below).

Supplemental Figure 7: Identification of non-IDH-mutant Glioma-CpG Island Methylator Phenotype (G-CIMP) tumor sample in glioblastoma cohort (n=30). **(A)** Heat map of 5-methylcytosine values and unsupervised hierarchical clustering of CpGs from G-CIMP genes identified in Noushmehr et al²². High levels of methylation, that is outlier DNA methylation, at these genes is suggestive of a G-CIMP phenotype for the sample in the far left portion of the dendrogram **(B)** Unsupervised hierarchical clustering of CpGs from G-CIMP genes in the TCGA glioblastoma data set (n = 154). High levels of methylation of G-CIMP genes is visualized in the far left branch of the dendrogram.

REFERENCES:

- 1 Kraus, T. F. *et al.* Loss of 5-hydroxymethylcytosine and intratumoral heterogeneity as an epigenomic hallmark of glioblastoma. *Tumour biology : the journal of the International Society for Oncodevelopmental Biology and Medicine* **36**, 8439-8446, doi:10.1007/s13277-015-3606-9 (2015).
- 2 Lian, C. G. *et al.* Loss of 5-hydroxymethylcytosine is an epigenetic hallmark of melanoma. *Cell* **150**, 1135-1146, doi:10.1016/j.cell.2012.07.033 (2012).
- 3 Christensen, B. C. *et al.* DNA methylation, isocitrate dehydrogenase mutation, and survival in glioma. *Journal of the National Cancer Institute* **103**, 143-153, doi:10.1093/jnci/djq497 (2011).
- 4 Nagarajan, R. P. *et al.* Recurrent epimutations activate gene body promoters in primary glioblastoma. *Genome research* **24**, 761-774, doi:10.1101/gr.164707.113 (2014).

- 5 Kozlenkov, A. *et al.* Differences in DNA methylation between human neuronal and glial cells are concentrated in enhancers and non-CpG sites. *Nucleic acids research* **42**, 109-127, doi:10.1093/nar/gkt838 (2014).
- 6 Kafer, G. R. *et al.* 5-Hydroxymethylcytosine Marks Sites of DNA Damage and Promotes Genome Stability. *Cell reports* **14**, 1283-1292, doi:10.1016/j.celrep.2016.01.035 (2016).
- 7 Brennan, C. W. *et al.* The somatic genomic landscape of glioblastoma. *Cell* **155**, 462-477, doi:10.1016/j.cell.2013.09.034 (2013).
- 8 Green, B. B. *et al.* Hydroxymethylation is uniquely distributed within term placenta, and is associated with gene expression. *FASEB journal : official publication of the Federation of American Societies for Experimental Biology*, doi:10.1096/fj.201600310R (2016).
- 9 Ivanov, M. *et al.* Single base resolution analysis of 5-hydroxymethylcytosine in 188 human genes: implications for hepatic gene expression. *Nucleic acids research*, doi:10.1093/nar/gkw316 (2016).
- 10 Uribe-Lewis, S. *et al.* 5-hydroxymethylcytosine marks promoters in colon that resist DNA hypermethylation in cancer. *Genome biology* **16**, 69, doi:10.1186/s13059-015-0605-5 (2015).
- 11 Stroud, H., Feng, S., Morey Kinney, S., Pradhan, S. & Jacobsen, S. E. 5-Hydroxymethylcytosine is associated with enhancers and gene bodies in human embryonic stem cells. *Genome biology* **12**, R54, doi:10.1186/gb-2011-12-6-r54 (2011).
- 12 Heyn, H. *et al.* Epigenomic analysis detects aberrant super-enhancer DNA methylation in human cancer. *Genome biology* **17**, 11, doi:10.1186/s13059-016-0879-2 (2016).
- 13 Suva, M. L. *et al.* Reconstructing and reprogramming the tumor-propagating potential of glioblastoma stem-like cells. *Cell* **157**, 580-594, doi:10.1016/j.cell.2014.02.030 (2014).
- 14 Orr, B. A., Haffner, M. C., Nelson, W. G., Yegnasubramanian, S. & Eberhart, C. G. Decreased 5-hydroxymethylcytosine is associated with neural progenitor phenotype in normal brain and shorter survival in malignant glioma. *PloS one* **7**, e41036, doi:10.1371/journal.pone.0041036 (2012).
- 15 Ostrom, Q. T. *et al.* The epidemiology of glioma in adults: a "state of the science" review. *Neuro-oncology* **16**, 896-913, doi:10.1093/neuonc/nou087 (2014).
- 16 Cohen, A. L., Holmen, S. L. & Colman, H. IDH1 and IDH2 mutations in gliomas. *Current neurology and neuroscience reports* **13**, 345, doi:10.1007/s11910-013-0345-4 (2013).
- 17 Feber, A. *et al.* Using high-density DNA methylation arrays to profile copy number alterations. *Genome biology* **15**, R30, doi:10.1186/gb-2014-15-2-r30 (2014).
- 18 Hatanpaa, K. J., Burma, S., Zhao, D. & Habib, A. A. Epidermal growth factor receptor in glioma: signal transduction, neuropathology, imaging, and radioresistance. *Neoplasia* **12**, 675-684 (2010).

- 19 Liu, F. *et al.* EGFR Mutation Promotes Glioblastoma through Epigenome and Transcription Factor Network Remodeling. *Molecular cell* **60**, 307-318, doi:10.1016/j.molcel.2015.09.002 (2015).
- 20 Forloni, M. *et al.* Oncogenic EGFR Represses the TET1 DNA Demethylase to Induce Silencing of Tumor Suppressors in Cancer Cells. *Cell reports*, doi:10.1016/j.celrep.2016.05.087.
- 21 Cancer Genome Atlas Research, N. *et al.* Comprehensive, Integrative Genomic Analysis of Diffuse Lower-Grade Gliomas. *The New England journal of medicine* **372**, 2481-2498, doi:10.1056/NEJMoa1402121 (2015).
- 22 Noushmehr, H. *et al.* Identification of a CpG island methylator phenotype that defines a distinct subgroup of glioma. *Cancer cell* **17**, 510-522, doi:10.1016/j.ccr.2010.03.017 (2010).
- 23 Cejas, P. *et al.* Chromatin immunoprecipitation from fixed clinical tissues reveals tumor-specific enhancer profiles. *Nature medicine* **22**, 685-691, doi:10.1038/nm.4085 (2016).
- 24 Heintzman, N. D. *et al.* Distinct and predictive chromatin signatures of transcriptional promoters and enhancers in the human genome. *Nature genetics* **39**, 311-318, doi:10.1038/ng1966 (2007).
- 25 Hnisz, D. *et al.* Super-enhancers in the control of cell identity and disease. *Cell* **155**, 934-947, doi:10.1016/j.cell.2013.09.053 (2013).
- 26 Ceccarelli, M. *et al.* Molecular Profiling Reveals Biologically Discrete Subsets and Pathways of Progression in Diffuse Glioma. *Cell* **164**, 550-563, doi:10.1016/j.cell.2015.12.028 (2016).

Reviewer #1 (Remarks to the Author)

The authors have adequately addressed my previous concerns. A notable issue that could be addressed including additional context and citations of prior work in 5hmC in glioma and other relevant biological systems. For example, They could cite a paper on 5-hydroxymethylcytosine in glioma (PMID: 24894482) to acknowledge prior work in glioma, as well as prior work in other systems (e.g. PMID: 26363184, PMID: 25263596, PMID: 22730288) that show 5hmC to be enriched in regulatory promoter and enhancer regions. Dysregulated DNA hydroxymethylation has also been described in AML (PMID: 25482556), a tumor that shares features with glioma in mutational profile (IDH).

Reviewer #2 (Remarks to the Author)

The authors have made a number of major revisions and included additional data/analysis. They have addressed prior critiques satisfactorily and I feel the paper is much improved.

Response to reviewer's comments on the manuscript: "5-Hydroxymethylcytosine localizes to enhancer elements and is associated with survival in patients with glioblastoma" (ORCID ID: 0000-0002-3385-0322)

Reviewer #1 (Remarks to the Author):

The authors have adequately addressed my previous concerns. A notable issue that could be addressed including additional context and citations of prior work in 5hmC in glioma and other relevant biological systems. For example, They could cite a paper on 5-hydroxymethylcytosine in glioma (PMID: 24894482) to acknowledge prior work in glioma, as well as prior work in other systems (e.g. PMID: 26363184, PMID: 25263596, PMID: 22730288) that show 5hmC to be enriched in regulatory promoter and enhancer regions. Dysregulated DNA hydroxymethylation has also been described in AML (PMID: 25482556), a tumor that shares features with glioma in mutational profile (IDH).

RESPONSE: As requested, we have now incorporated the suggested citations highlighting prior work into the revised version of the manuscript. Specifically, we now include appropriate citation of previous glioma 5hmC studies on p. 9 and mention the previous work that reported dysregulated 5hmC in AML on p. 10. Additionally, we added the following text on p. 10 to acknowledge prior work on the enrichment of 5hmC in regulatory regions in papers investigating cellular differentiation:

"Prior work has demonstrated that 5hmC modulates enhancer activity and regulates gene expression programs during cellular differentiation suggesting that 5hmC deregulation may impact the dedifferentiation observed in glioblastoma^{1, 2, 3, 4}."

Reviewer #2 (Remarks to the Author):

The authors have made a number of major revisions and included additional data/analysis. They have addressed prior critiques satisfactorily and I feel the paper is much improved.

RESPONSE: We thank the reviewer for their positive feedback and for their comments that have strengthened the manuscript.

REFERENCES:

1. Friedmann-Morvinski D, *et al.* Dedifferentiation of neurons and astrocytes by oncogenes can induce gliomas in mice. *Science* **338**, 1080-1084 (2012).

2. Rampal R, *et al.* DNA hydroxymethylation profiling reveals that WT1 mutations result in loss of TET2 function in acute myeloid leukemia. *Cell reports* **9**, 1841-1855 (2014).
3. Hon GC, *et al.* 5mC oxidation by Tet2 modulates enhancer activity and timing of transcriptome reprogramming during differentiation. *Molecular cell* **56**, 286-297 (2014).
4. Taylor SE, Li YH, Smeriglio P, Rath M, Wong WH, Bhutani N. Stable 5-Hydroxymethylcytosine (5hmC) Acquisition Marks Gene Activation During Chondrogenic Differentiation. *Journal of bone and mineral research : the official journal of the American Society for Bone and Mineral Research* **31**, 524-534 (2016).